# Nucleolin Therapeutic Targeting Decreases Pancreatic Cancer Immunosuppression

**DOI:** 10.3390/cancers14174265

**Published:** 2022-08-31

**Authors:** Matteo Ponzo, Anais Debesset, Mélissande Cossutta, Mounira Chalabi-Dchar, Claire Houppe, Caroline Pilon, Alba Nicolas-Boluda, Sylvain Meunier, Fabio Raineri, Allan Thiolat, Rémy Nicolle, Federica Maione, Serena Brundu, Carina Florina Cojocaru, Philippe Bouvet, Corinne Bousquet, Florence Gazeau, Christophe Tournigand, José Courty, Enrico Giraudo, José L. Cohen, Ilaria Cascone

**Affiliations:** 1Immune Regulation and Biotherapy, Inserm U955, IMRB University of Paris-Est Creteil (UPEC) 8, INSERM, IMRB, F-94010 Créteil, France; ponzo.matteo88@gmail.com (M.P.); anais.debesset@inserm.fr (A.D.); mc@imstarsa.com (M.C.); claire.justine@u-pec.fr (C.H.); caroline.pilon@inserm.fr (C.P.); sylvain.meunier@inserm.fr (S.M.); fabio.raineri@inserm.fr (F.R.); allan.thiolat@inserm.fr (A.T.); christophe.tournigand@aphp.fr (C.T.); courty@u-pec.fr (J.C.); jose.cohen@inserm.fr (J.L.C.); 2Cancer Research Center of Lyon, Cancer Cell Plasticity Department, University of Lyon, UMR INSERM 1052 CNRS 5286, Centre Léon Bérard, F-69008 Lyon, France; mounira.chalabi@lyon.unicancer.fr; 3AP-HP, Groupe Hospitalo-Universitaire Chenevier Mondor, Centre D’investigation Clinique Biothérapie, F-94010 Créteil, France; florence.gazeau@univ-paris-diderot.fr; 4Matières et Systèmes Complexes (MSC), Université de Paris, CNRS UMR 7057, F-75006 Paris, France; alba.nicolas.boluda@gmail.com (A.N.-B.); philippe.bouvet@ens-lyon.fr (P.B.); 5Programme Cartes d’Identité des Tumeurs (CIT), Ligue Nationale Contre le Cancer, F-75013 Paris, France; remy.nicolle@inserm.fr; 6Laboratory of Tumor Microenvironment, Candiolo Cancer Institute, FPO-IRCCS, 10060 Candiolo, Italy; federica.maione@ircc.it (F.M.); serena.brundu@ircc.it (S.B.); carina.cojocaru@ircc.it (C.F.C.); enrico.giraudo@ircc.it (E.G.); 7Department of Science and Drug Technology, University of Torino, 10125 Torino, Italy; 8Ecole Normale Supérieure de Lyon, University of Lyon, F-69342 Lyon, France; 9UMR INSERM-1037, Cancer Research Center of Toulouse (CRCT), Toulouse University III, F-31037 Toulouse, France; corinne.bousquet@inserm.fr; 10AP-HP, Service d’Oncologie Médicale, Groupe Hospitalo-Universitaire Chenevier Mondor, F-94010 Créteil, France

**Keywords:** vessels, nucleolin, immune cells, PDAC

## Abstract

**Simple Summary:**

Nucleolin (NCL) regulates tumour growth and angiogenesis, and its inhibition normalizes tumour vessels and impairs pancreatic ductal adenocarcinoma (PDAC) growth. Since tumour vessel normalization promotes immunostimulatory reprogramming, we investigated the effects of a selective inhibitor of NCL, the pseudopeptide N6L, on the immune microenvironment of PDAC. This work highlights a new therapeutic strategy that restrains immunosuppressive cells to promote T- cell recruitment and activation and to re-program the tumour stroma of PDAC.

**Abstract:**

Background: The pancreatic ductal adenocarcinoma (PDAC) microenvironment is highly fibrotic and hypoxic, with poor immune cell infiltration. Recently, we showed that nucleolin (NCL) inhibition normalizes tumour vessels and impairs PDAC growth. Methods: Immunocompetent mouse models of PDAC were treated by the pseudopeptide N6L, which selectively inhibits NCL. Tumour-infiltrating immune cells and changes in the tumour microenvironment were analysed. Results: N6L reduced the proportion of regulatory T cells (Tregs) and myeloid-derived suppressor cells (MDSCs) and increased tumour-infiltrated T lymphocytes (TILs) with an activated phenotype. Low-dose anti-VEGFR2 treatment normalized PDAC vessels but did not modulate the immune suppressive microenvironment. RNAseq analysis of N6L-treated PDAC tumours revealed a reduction of cancer-associated fibroblast (CAF) expansion in vivo and in vitro. Notably, N6L treatment decreased IL-6 levels both in tumour tissues and in serum. Treating mPDAC by an antibody blocking IL-6 reduced the proportion of Tregs and MDSCs and increased the amount of TILs, thus mimicking the effects of N6L. Conclusions: These results demonstrate that NCL inhibition blocks the amplification of lymphoid and myeloid immunosuppressive cells and promotes T cell activation in PDAC through a new mechanism of action dependent on the direct inhibition of the tumoral stroma.

## 1. Introduction

Pancreatic ductal adenocarcinoma (PDAC) comprises 2.5% of worldwide cancer cases [1] and is the fourth leading cancer in Europe [2]. Surgical resection is the sole curable approach for localized PDAC, but only 20% of tumours are resectable at the time of diagnosis, and 80% of patients have locally advanced or metastatic disease [3]. Current chemotherapeutic regimens are based on 5-fluorouracil or gemcitabine and only offer survival times in the range of months in the palliative setting; there are effectively no FDA-approved targeted therapeutics for PDAC [4].

The malignancy of PDAC is partly due both to the intrinsic resistance of cancer cells to chemotherapy and to the tumour microenvironment [5]. The stroma, predominant in the tumour mass is constituted by non-neoplastic cells and a highly dense and fibrotic matrix. One of the main non-neoplastic types of PDAC stromal cells is cancer-associated fibroblasts (CAFs), producing the extracellular matrix and secreting growth factors and cytokines that promote tumour progression and chemoresistance [6,7]. Analysis of the expression profile of tumours from patients describes the presence of a normal and an activated signature of CAFs, with the latter being associated with a poorer prognosis [8,9,10,11,12]. PDAC stroma can also be infiltrated by T and myeloid cell populations [13,14], and patients with the higher proportions of CD8^+^ and CD4^+^ T cells together with dendritic cells have improved prognosis [15]. T cells are predominantly antigen-experienced effector memory T cells, with the potential to activate immune response [16,17]. However, the PDAC microenvironment contains also multiple immunosuppressive cells, such as regulatory T cells (Tregs) and myeloid-derived suppressor cells (MDSCs), that inhibit T cell activation and are associated with tumour-permissive anergy and poor prognosis [18,19]. Another peculiarity of the PDAC microenvironment is the tumour vascularization. Blood vessels in PDAC are compressed by the fibrotic stroma; they have an aberrant structure and the vessel network is chaotic, resulting in local hypoxia, inefficient drug delivery and poor immune cell infiltration [20,21], all concomitantly leading to tumour progression. 

A growing body of evidence from our laboratory and other groups has demonstrated that NCL regulates tumour growth and angiogenesis [22,23,24,25,26,27]. NCL is a multifunctional protein overexpressed in different types of cancer, including PDAC [24,28]. In addition, we observed that NCL was a significant prognostic factor in human PDAC patients, [24]. NCL is present on the surface of many cell types and is a hallmark of proliferative and cancer cells [29,30]. We developed a multivalent pseudopeptide N6L able to bind to and inhibit NCL [27], which exerted promising antitumour activity in prostate cancers, melanoma, lymphoma and other cancers [24,27,31,32,33,34]. N6L is constituted by a lysine-rich template of a 3_10_ helical matrix composed of six repeats of Lys-Aib-Gly, the pseudotripeptides Lysψ[CH_2_N]Pro-Arg being grafted onto the *ϵ* NH_2_ of the matrix Lys residues [27]. Importantly, N6L not only blocks tumour growth and distal metastasis but also normalizes tumour vasculature, leading to decreased tumour hypoxia and improved delivery and efficacy of chemotherapy in orthotopic PDAC mouse models [24,35]. Several anti-angiogenic molecules have been developed in the past years to stifle tumour growth. It is now assumed that tumour vessel normalization strategies, by restoring blood vessel structure and functions, are able to reduce tumour hypoxia and favour CD8 T cell infiltration and activation [36,37,38]. Since N6L induced tumour vessel normalization, we sought to investigate whether N6L was also able to influence the landscape of the anti-tumour immune response. We demonstrate that N6L remodels the PDAC tumour microenvironment, decreasing the frequency of immunosuppressive cells by targeting the tumour stroma and IL6 production by CAFs.

## 2. Materials and Methods

### 2.1. Cell Culture

Murine pancreatic cancer cells, mPDAC cells, were isolated and validated, as previously described [24], from tumour-bearing *p48cre, Kras^LSL_G12D^, p53^R172H/+^, Ink4a/Arf^flox/+^* FVB/n mice and KPC cells from tumour-bearing *p48cre, Kras^LSL_G12D^, p53^R172H/+^* C57Bl/6 mice (kindly provided and validated by D. Saur, TranslaTUM, Munich, Germany). mPDAC, KPC, HEK293T (293T), MIA PaCa-2, and PANC-1 cells were cultured in DMEM (41965-039, Gibco) 10% FBS as previously described [24]. Immortalized human CAFs were cultured in in Advanced DMEM/F12 (Thermo 12634) 10% FBS at 37 °C, 5% CO_2_, as previously described [12].

### 2.2. Incucyte^®^ Live-Cell Imaging Analysis

CAFs (10^4^) were cultured in seeded in 96-well plates and allowed to adhere overnight. Cells were treated with 1–5µM of N6L for 96 h. Cell growth was assessed by IncuCyte ZOOM^®^ live-cell imaging to measure cell confluence after N6L treatment. IncuCyte cell recognition software calculated values based on percentage of cell confluence over time (Essen Biosciences, Ann Arbor, MI, USA).

### 2.3. Tumour Mouse Model

Tumour mouse models were obtained by orthotopic injection of murine PDAC cells into syngenic FVB/n (for mPDAC mice) or C57BL/6 mice (for KPC mice) obtained from Janvier Labs. Ten weeks of age FVB/n mice (weight average of 20 g) were injected orthotopically in the pancreas with mPDAC cells (10^3^ cells/mouse in 50 μL) as previously described [24,39], and C57BL/6 mice were injected with KPC cells (10^4^ cells/mouse). Previously, we defined as a starting point to perform a regression trial one week after cancer cell inoculation, the time period in which tumours reached a volume of approximately 80 mm^3^ [24,39]. The treatment started after one week from cell injection. Both mPDAC and KPC mice were treated 3 times a week with either N6L (7 mg/kg) or a vehicle (saline solution) as a control. mPDAC mice were sacrificed at day 21 and KPC at day 35 as the fixed endpoint of the experiments. For anti-IL6 experiments, mPDAC mice were treated 3 times a week for 2 weeks by i.p. injections with either anti-IL6 antibody (BioXcell ref. BE0046 clone: MP5-20F3, 200 µg/mouse) or isotype control (rat IgG1 isotype control, anti-horseradish peroxidase ref. BE0088 clone: HRPN, 200 µg/mouse). For anti-VEGR2 experiments, mice were treated as previously indicated [40], 2 times for a week for 2 weeks by i.p. injections with either anti-VEGFR2 antibody (BioXcell ref.BE0060, clone:DC101, 20 mg/kg) or isotype control (rat IgG1 isotype control, anti-horseradish peroxidase ref. BE0088 clone: HRPN, 20 mg/kg). Mice were sacrificed, and total tumour burden was quantified as previously described [24]. Tumours, spleens, and pancreatic or axillary lymph nodes were collected. *For measuring tumour stiffness,* shear wave elastography (SWE) measurements were performed every 3–4 days during the entire follow-up of the tumour growth, as previously described [41]. Images were acquired with the ultrasound device Aixplorer (SuperSonic Imagine, Aix-en-Provence, France) using a 15-MHz superficial probe dedicated to research (256 elements, 0.125 µm pitch). SWE images were also analysed using an in-house MATLAB code to recover the stiffness map. 

### 2.4. Western Blot Analysis and Protein Array

Twenty micrograms of total protein lysates were run on a 10% SDS polyacrylamide gel and transferred onto a nitrocellulose member (Bio-Rad, Hercules, CA, USA). The membrane was blocked with 3% non-fat milk in TBS Tween 0.1% (TBST) incubated with anti-nucleolin (Abcam, ab22758, 1:1000, Cambridge, UK) and anti-β-actin (Sigma-Aldrich, A3854, clone AC-15, St. Louis, MO, USA). Proteins were detected by chemiluminescence using an anti-rabbit or anti-mouse peroxidase-conjugated antibody (Cell Signalling) diluted 1:10,000, and Clarity ECL substrate (Bio-Rad). Images were acquired with a ChemiDoc XRS+ (Bio-Rad). For IL-6 analysis in tumours, tumours were harvested in PBS 1X and cut in small fragments. Fragments were incubated 2 h at room temperature under agitation. Supernatants and fragments were separated, and supernatants were frozen and the amount of IL-6 was analysed by ELISA (R&D Systems, Minneapolis, MN, USA). For IL-6 analysis in plasma, pools of plasma from healthy, tumour-bearing mice treated by saline solution or tumour-bearing mice treated by N6L were analysed by using a protein array (XL cytokine array kit, R&D Systems #ARY028).

### 2.5. Flow Cytometry Analysis

Pancreatic tumours and axillary lymph nodes were mechanically dissociated and counted. Mouse spleens were mechanically dissociated then passed through 70 μm cell strainers (Falcon 352350). Cell suspension was incubated for 5 min in ACK buffer (1.5 M NH4CL, 100 mM KHCO3, 10 mM EDTA) in order to eliminate erythrocytes, and 1 × 10^6^ cells were used for staining. mPDAC tumours were put in RPMI-1640 + 2% FBS + collagenase IV (1.5 mg/mL; #WOLS04186 Worthington Biochemical) + DNase I (0.1 mg/mL; Sigma-Aldrich, #D5025-150KU), minced with scissors to sub-millimetre pieces, and incubated at 37 °C for 60 min in rotation. Samples were passed through 70 μm cell strainers (Falcon 352350) and centrifuged at 300× *g* for 5 min. Then, cell suspension was centrifuged in a gradient of Ficoll (# 17144003, GE Healthcare) at 400× *g* without brakes for 40 min. Immune cells were harvested, washed, and counted. Then, 1 × 10^6^ cells from indicated tissues were incubated with an anti-CD16/CD32 mAb (#130-07-594, Miltenyi) to block FcγRIII/II at 4 °C for 30 min. Living cells were incubated with extracellular staining antibodies (CD45-ef450 eBioscience #48-0451-82 1:40, CD4-FITC Miltenyi #130-118-692 1:100, CD8-PE eBioscience #12-0081-82 1:80, PD1-PerCP-efluor710 eBioscience #46-9985-82 1:160, CD25-PE-Cy7 eBioscience #25-0251-82 1:160, Ly6C-PE BD Biosciences #560592 1:100, Ly6G-Brillant violet 421 BD Biosciences #562737 1:100 CD11b-PE-Cy7 eBioscience #25-0112-82 1:160, F4/80-FITC Biolegend #123-120 1:80, CD11c-PE-Cy5.5 eBioscience #35-0114-82 1:80, BD Horizon™ Fixable Viability Stain 510 (FVS510) 1:500) at 4 °C for 30 min. After incubation, cells were washed and fixed with 100 μL of FoxP3/transcription Kit Fix/perm (Invitrogen #00-552300) at 4 °C for 1 h or overnight and washed twice with Perm Wash (Invitrogen, #00-552300). Then, cells were incubated at 4 °C for 30 min with the following antibodies to obtain an intracellular staining: FoxP3-eFluor660 eBioscience #50-5773-82 1:100, IFN-γ-APC Miltenyi #130-109-770 1:50, MRC-AlexaFluor 647 BD Biosciences #565250 1:50. For IFN-γ staining, single cell suspension was incubated in RPMI 10% FBS + PMA (Sigma-Aldrich, 1 μg/mL) + ionomycin (Sigma-Aldrich, 0.5 μg/mL) + Golgi Plug (Cytofix/Cytoperm TM Plus, BD Bioscience, #555028 1:1000) at 37 °C for 4 h in order to block exocytosis from Golgi apparatus and to stimulate cytokine production. Then, cells were washed with PBS + 3% FBS and stained as described above. All samples were analysed using a Canto II flow cytometer (BD Biosciences). Compensation was performed using FMO (fluorescence minus one) controls. All cytometry data were analysed using FlowJo software.

### 2.6. Immunohistochemistry and Immunofluorescence of Tumour Tissues

Pancreatic tumours were paraffin embedded. Then, 5 μm slices of tumour tissues were incubated in Target Retrieval Solution Citrate pH 6 (Dako, #S2369, Agilent, Santa Clara, CA, USA) at 95 °C in a hot water bath for 40 min to remove paraffin and to unmask antigens. After 20 min of cooling, sections were incubated in Peroxidase Blocking Reagent (Dako) at room temperature for 10 min to block endogenous peroxidase activities. Nonspecific antibody binding was prevented using Protein Block Solution (Dako, #X0909) at room temperature for 10 min. Primary antibodies, anti-CD8^+^ (Cell signalling, #98941 1:100), or anti-αSMA (Abcam, #ab5694 1:100) were incubated at room temperature for 1 h in blocking buffer (Dako). Then, staining was performed using the DAKO EnVision+ System and Peroxidase following the manufacturer’s instructions. Tumour sections were counterstained using hematoxylin (Roth). The number of CD8^+^ cells was analysed by QuPath software, by manually counting the cells infiltrated in the tumours in at least 4 regions of each tumour.

### 2.7. RNA Extraction, Reverse-Transcription qPCR, and Transcriptome Analysis

Total RNA of cells or tumours was prepared by TRIzol (Invitrogen) extraction. For qPCR, total RNA was reverse-transcribed using hexamer random primers and a first-strand cDNA synthesis kit (Fermentas), and the synthesized cDNA was used for RT–qPCR using FastStart Universal SYBR Green Master (ROX) (Roche). The primers used for qPCR were hIL-6 (Fw: 5′AATTCGGTACATCCTCGACGG3′, Rv: 5′GGTTGTTTTCTGCCAGTGCC3′) and b-ACTIN (Fw: 5′GTTACAGGAAGTCCCTTGCCATCC3′; Rev: 5′CACCTCCCCTGTGTGGACTTGGG3′). RNA extracted from Control (*n* = 3) or N6L-treated bulk tumours (*n* = 3) were sequenced on the Illumina NextSeq500 at the Nice-Sophia-Antipolis Functional Genomics Platform. The obtained libraries of sequences (reads) were aligned with STAR on the mm10 genome version during the primary analysis. Results were normalized and analysed for differences by log2FoldChange between the Control and N6L-treated tumours using the DESeq2 package in R for the global analysis (size factors normalization). The *p*-value after adjustment for multiple testing was applied to the log2FoldChange to discriminate the genes differentially expressed. MCPcounter [42] was applied to the normalized gene expression value to estimate the quantification of fibroblasts. Functional enrichment analyses were performed by GenoBiToUs (University of Turin) using the ClusterProfiler (Yu 2012) package. Gene ontology enrichment *p*-values of the differentially expressed genes (Figure 4B) were obtained with an exact Fisher test. Pre-ranked Gene Set Enrichment Analysis was performed by ranking genes based on their differential expression fold change between the N6L treatment and control. The stroma signatures, namely activated stroma and inflammatory stroma, were taken from [9].

### 2.8. Study Approval

All in vivo experiments were carried out with the approval of the Institutional Ethical Committees and of the Italian and French Ministries of Health in compliance with European laws and policies. 

### 2.9. Statistical Analysis

Statistical analyses were performed using GraphPad Prism software. Unless indicated otherwise, bars represent mean ± Standard Error of Mean (S.E.M.). For continuous variables, we first tested both normality and equal variance. For two-group comparisons, we analysed the data using the two-tailed Student’s *t*-test when the data passed both tests or the Mann-Whitney U-test if both tests failed. Test results are reported in the figure legends.

## 3. Results

### 3.1. N6L Decreases Immunosuppressive Cell Infiltration into the Tumour Microenvironment of the mPDAC Mouse Models

We previously demonstrated that the orthotopic PDAC tumours (mPDAC) reproduced human tumour histopathology with a fibrotic stroma and a strong hypoxia [24]. Here, we aimed to characterize the immune microenvironment of the orthotopic mPDAC tumours. Immunohistochemistry analysis revealed that the tumour microenvironment of mPDAC was infiltrated by CD8 T-cells (Appendix A), among which 2.9% of cells were CD8^+^ and 21.7% were CD4^+^ (Appendix A). The remaining 55.5% of CD45^+^ cells were mainly myeloid CD11b^+^ cells (Appendix A). Within the CD4^+^ cells, 26.9% were CD4^+^FOXP3^+^ regulatory T cells (Tregs) (Appendix A). 

When we studied whether the presence of pancreatic tumours could modify the immune cell content of the pancreatic draining lymph nodes (LNs), we did not observe major modifications regarding the percentages of CD45^+^CD8^+^ cells and CD45^+^CD4^+^ cells (Appendix A). However, compared to healthy mice, the proportion of Treg cells significantly increased in tumour-bearing mice, leading to a 2.3-fold decrease of the CD8^+^/Treg ratio (Appendix A). Thus, mPDAC mice not only reproduced the histopathological, fibrotic, and hypoxic characteristics of human PDACs, but also an immunosuppressive environment characterized by Treg enrichment and a reduced amount of cytotoxic CD8^+^ T cells in tumour-draining lymph nodes.

We initially demonstrated that N6L decreases tumour growth and induces tumour vessel normalization in our model of orthotopic PDAC [24]. Herein, we aimed to assess whether N6L treatment could affect immune cell infiltration into the tumours. In experimental settings in which we reproduced the published protocol of treatment and the therapeutic effect of N6L on tumour growth (Figure 1A) [24], immune cells from tumours were isolated and analysed by flow cytometry. The relative proportions of tumour CD4^+^ and CD8^+^ T cells (Figure 1B) were not modified by N6L treatment. However, Tregs were reduced by a factor of two (Figure 1C), leading to a statistically significant increase of the CD8^+^/Treg ratio (Figure 1C) in the tumours of N6L-treated mice compared with controls. Treg content and CD8/Treg ratio were not modified in tumour lymph nodes or spleen of N6L-treated mice compared with controls (Figure 1D), suggesting a mainly localized effect of N6L on immune cells into the tumour microenvironment. 

PDAC are described as highly infiltrated by myeloid cells, whose subpopulations could have immunosuppressive functions [43,44,45]. In mouse PDAC, we observed that the proportion of total myeloid cells (CD11b^+^) within the population of CD45^+^ cells decreased in N6L-treated tumour-bearing mice compared to control mice (Figure 2A), whereas macrophages (CD11b^+^/F480^+^) and the frequency of M1 (CD11b^+^F4/80^+^MRC^-^CD11c^+^) and M2 (CD11b^+^F4/80^+^MRC^+^CD11c^−^) polarized macrophages were not (Figure 2B). PDAC myeloid-derived suppressor cells (MDSCs) were analysed as previously described [43,45,46]. Granulocytic myeloid-derived suppressor cells (Gr-MDSCs) (CD11b^+^Ly6G^+^Ly6C^low^) decreased by 2.5-fold in N6L-treated tumours compared to control tumours (Figure 2C), and the monocytic myeloid-derived suppressor cell (M-MDSCs) population (CD11b^+^Ly6G^low^Ly6C^+^) slightly increased (Figure 2C). 

Thus, we demonstrate that N6L treatment in PDAC-bearing mice reduced the proportion of two main immunosuppressive cell populations, i.e., Tregs and Gr-MDSCs, within the tumours.

### 3.2. N6L Increases Infiltration and Activation of T lymphocytes into PDAC Microenvironment

The CD8^+^ T cell proportion did not change after N6L treatment (Figure 1B). However, the analysis by FACS did not allow separating peritumoral and tumoral regions. We therefore explored the impact of N6L on CD8^+^ T cell infiltration within the tumoral region by immunohistochemistry (Figure 3A, arrows). Tumoral regions presenting typical cancerous ducts and neoplasia have been identified in hematoxylin eosin (H&E) stains for each tumour. Adjacent slides were immunostained for CD8 detection by using an anti-CD8 antibody. The number of CD8^+^ T cells present in the tumoral regions (Figure 3A) was counted by the QuPath software and plotted in Figure 3B. While the number of tumour-infiltrated CD8^+^ T cells was very low in control tumours, it dramatically increased in N6L-treated mice (Figure 3A arrows, and Figure 3B). Interestingly, this was associated with an increased frequency of interferon-γ (IFN-γ)-producing effector T cells infiltrated in tumours of N6L-treated mice (Figure 3C). In order to consolidate these observations, we reproduced experiments using a second orthotopic PDAC model consisting of injecting KPC cells in C57BL/6 mice. Since KPC growth is slower than that of mPDAC, mice were treated for 4 weeks and sacrificed at day 35. Similarly to mPDAC, N6L reduced the KPC tumour growth (Appendix A) and increased the number of tumour-infiltrated CD8^+^ T cells in tumours (Appendix A). 

### 3.3. N6L Impacts Tumour Stroma and Inhibits IL-6 Production

To understand the impact of N6L on the tumour microenvironment we performed a RNA-Seq analysis to assess the gene expression profile of N6L-treated versus control tumours (Figure 4). The heat map of the statistically modulated genes between control and N6L-treated mice is shown in Figure 4A. A total of 59 genes were significantly downregulated and 33 upregulated (Appendix A, respectively). GO analysis of the 59 downregulated genes shows an enrichment of biological processes of angiogenesis, vessel sprouting, and vessel morphogenesis (Figure 4B). Moreover, GSEA analysis of all regulated genes showed a significant downregulation of cell proliferation and angiogenesis processes (Figure 4C). These results are coherent with the effects of N6L in decreasing cell proliferation of PDAC and angiogenesis [24,26]. 

To understand whether the normalization of the tumour vasculature had some impact on the tumour immune cell infiltration of mPDAC, we treated mice with a low dose of anti-VEGFR2. A low dose of anti-VEGFR2 induces and sustains vessel normalization [40] and enhances the coverage of tumour blood vessels by the NG2^+^ and PDGR^+^ pericytes (Appendix A). Indeed, low-dose anti-VEGFR2 did not impact tumour volume (Appendix A) and did not change the frequency of Treg and Gr-MDSCs (Appendix A). Thus, while both N6L and anti-VEGFR2 normalizes tumour vessels in mPDAC, only N6L has the capacity to re-program the immune cell environment. 

When looking at the GO analysis of N6L downregulated genes (Figure 4B), biological processes linked to smooth muscle cell proliferation appeared as statistically regulated. Using the microenvironment cell populations (MCP)-counter, a transcriptome-based computational method, we observed that the number of fibroblasts [42] was decreased in mPDAC tumours treated by N6L (Figure 4D). In parallel, GSEA analysis of all modulated genes revealed a significant downregulation of the gene signatures corresponding to PDAC-activated stroma and PDAC-activated inflammatory stroma (Figure 4E), related to myofibroblastic CAFs (myCAFs) and inflammatory CAFs (iCAFs) established by the stroma signatures previously described [9]. The inflammatory stroma component was depicted as associated with high expression levels of IL-6 [9]. In our experiments, IL-6 was one of the top downregulated genes by N6L (Appendix A), and IL-6 production was a biological process downregulated by N6L in GSEA analysis (Figure 4F). The protein level of IL-6 was analysed in the tumour (Figure 4G) and in the plasma (Figure 4H) of mice treated or not by N6L. The supernatant of collected tumours was analysed by ELISA, and the amount of IL-6 was normalized by the tumour volume and expressed as pg of IL6 in mm^2^ of the tumour. Notably, the amount of IL-6 was significantly reduced in N6L-treated tumours compared to controls (Figure 4G). A pool of plasma from seven healthy mice, seven tumour-bearing mice treated by saline solution, or seven tumour-bearing mice treated by N6L was analysed by using a protein array (Figure 4H). The semi-quantitative analysis of IL-6 is represented as the results from the pool of each cohort of seven mice, calculated as fold change (Figure 4H). We observed elevated IL-6 levels in the plasma of tumour-bearing mice compared to healthy mice, confirming its overexpression during tumour progression (Figure 4H). Importantly, IL-6 level decreased in the plasma of N6L-treated mice compared to control tumour-bearing mice (Figure 4H).

Stemming from these observations, we evaluated the impact of N6L on the area of CAFs in mPDAC tumours by an anti-α-SMA staining and observed that α-SMA was strongly reduced in N6L-treated tumours compared with control mice (Figure 4I,J). Since CAF-dependent fibrosis contributes to tumour stiffness, the tumour stiffness of mPDAC was measured by shear wave elastography (SWE) as a marker of activated stroma. Tumour stiffness was significantly decreased in N6L-treated tumours compared to control tumours (Figure 4K,L). Together, these results suggested that N6L targets the activation of the tumour stroma.

### 3.4. IL-6 Blockade Mimics N6L Effects on PDAC Immune Microenvironment

We sought to evaluate the direct effect of N6L treatment on CAFs and used human immortalized CAFs. Firstly, we verified that nucleolin was expressed by human CAFs compared to PANC-1 and MIA PaCa-2 tumour cell lines (Figure 5A,B). Interestingly, treatment of CAFs by increasing concentrations of N6L decreased nucleolin protein levels starting at 5 μM (Figure 5C,D), similar to tumour cells previously described [24]. Increasing N6L doses impaired CAF proliferation (Figure 5E,F). IL-6 mRNA level was reduced in CAFs treated or not by N6L at 5 μM after 72 h (Figure 5G).

Based on these data, we sought to investigate the functional role of an IL-6 blockade on the tumour microenvironment of mPDAC models. For this, tumour-bearing mice were treated with 200 µg of anti-IL-6 blocking antibody with the same therapeutic scheme as used for N6L. Notably, even though anti-IL6 treatment did not influence tumour volume (Figure 6A), the percentage of Tregs (Figure 6B) and Gr-MDSCs (Figure 6C) decreased by a factor of two, and the CD8/Tregs ratio increased in anti-IL-6-treated tumours, compared to control tumours (Figure 6D). Similarly to N6L, the number of infiltrated CD8^+^ T cells within the tumour increased in anti-IL6-treated tumour sections compared to the control (Figure 6E,F). However, the frequency of interferon-γ (IFN-γ)-producing effector T cells infiltrated in tumours of anti-IL-6 treated mice did not vary (Figure 6G). Thus, anti-IL-6 treatment mimics the effect of N6L by decreasing immunosuppressive cells in the tumour microenvironment and increasing CD8^+^ tumour infiltration.

## 4. Discussion

In this study, we describe for the first time that N6L, a nucleolin-targeting treatment could modify the immune landscape in a mouse model of PDAC by decreasing immune-suppressive cells and promoting CD8^+^ T cell infiltration and activation.

The immunological classification of cancers in relation to disease severity and patient prognosis has been the subject of intense investigation in recent years. In human PDAC, some immunophenotypic classification correlates with a prognostic significance [9,47,48]. Briefly, the number of Tregs infiltrated into the human PDAC microenvironment increases in invasive ductal carcinoma compared to low-grade PanIN, while the CD8^+^ T cell number decreases [48]. High Treg prevalence in CD4^+^ T cells and low CD8^+^ T cell density are independently correlated with advanced tumour stages, high tumour grade, and distant metastasis [48,49]. By analysing the level of effector and regulatory T cell infiltration, Wartenberg et al. established three types of immunophenotypic classification of PDAC [47]. Poor T infiltration and high Treg enrichment have been described as the “immune escape” phenotype and can affect prognosis of PDAC [47,48,49]. Because the immunophenotype of PDAC tumours seems to be associated with some clinical characteristics, we sought to carefully characterize the tumour microenvironment of PDAC mouse models. We showed that CD8^+^ cells poorly infiltrate mPDAC tumours, while the frequency of Tregs is very high, resulting in a very low CD8/Treg ratio. Total myeloid CD11b+ cells were also elevated in mPDAC tumours, as described in both human and murine tumours [45,50]. Tregs should expand in tumour-draining lymph nodes [51,52], and we observed that tumour-draining lymph nodes of mPDAC models had higher Treg frequency and lower CD8/Treg ratio than healthy pancreatic lymph nodes. Moreover, Treg frequency of mPDAC tumours was much higher than in draining lymph nodes with a poorer CD8/Treg ratio. These results together with the high infiltration of total myeloid CD11b^+^ cells suggest that the immune microenvironment of mPDAC orthotopic tumours has an immune-escape profile and that it accurately mimics the human tumours.

As observed in humans in many cancers, we hypothesized that a more favourable evolution of PDACs could be associated with a modification or even a strengthening of their immune environment. Indeed, we unveiled a novel effect of N6L, since in tumour-bearing mice treated with N6L, the frequency of Tregs was reduced by half and the ratio CD8/Treg increased by 2.7-fold. Importantly, the Treg decrease was associated with higher CD8^+^ T cell infiltration into the tumours and a three-fold higher proportion of activated IFNγ^+^-producing T cells after N6L treatment. However, Treg frequency and the CD8/Treg ratio of lymph nodes and spleens did not change. This result supports that N6L specifically targets the tumour immune microenvironment.

In addition, PDAC are generally highly infiltrated by immunosuppressive myeloid cells that drive T cell exclusion and dysfunction [53,54,55]. Inflammatory monocyte mobilization decreases patient survival, and targeting the CCL2/CCR2 or CSF1/CSF1R axis that allows targeting macrophages or myeloid cell lineages improves chemotherapeutic efficacy and anti-tumour T-cell response [44,54,56]. Moreover, agonists of CD11b^+^ cells reprogram the remaining macrophages and decrease tumour growth [44,55]. In our mouse model of orthotopic PDAC, we observed that N6L treatment affects the relative abundance of the total myeloid (CD11b^+^) cells but does not significantly change the proportion of the tumour-associated macrophages (TAMs) (CD11b^+^/F4/80^+^); neither M1 or M2 polarized TAMs. We also could detect Gr-MDSCs and M-MDSCs populations. In mice, Gr-MDSCs are recruited to primary tumours by tumour cells, suppress proliferation, and induce apoptosis of T cells [43]. The specific targeting of the Gr-MDSCs by using an antibody anti-LY6G decreases Gr-MDSCs in PDAC tissues and promotes T cell activation and proliferation [43]. Interestingly, Gr-MDSCs depletion was accompanied by an increase in the number of M-MDSCs, and authors have suggested a homeostatic regulation between the two MDSCs subpopulations [43]. In our hands, N6L treatment decreased Gr-MDSCs together with an increase in the proportion of the M-MDSCs. However, contrary to the anti-LY6G, only N6L treatment was also able to reduce tumour growth and promote T cell infiltration and activation, suggesting that solely targeting MDSCs is not sufficient to mediate an anti-tumour effect in PDAC.

Whether infiltration and expansion of Tregs and MDSCs instil an immunosuppressive microenvironment, it becomes clearer that stromal cells participate in this regulation [5,57]. CAF density supports a fibrotic stroma, which contributes to the hypoxia of the tumour via the compression of tumour vessels and the interstitial pressure. In turn, the fibrotic stroma increases tumour stiffening, which restrains T cell migration [41,58]. In hypoxic conditions, CAFs are activated and express inflammatory and immunosuppressive factors, such as IL-6 and TGFβ, which in turn recruit suppressive Tregs, myeloid cells, and M2 macrophages [5,19]. Moreover, a pharmacological normalization of the CAFs secretome in PDAC remodel the immunosuppressive myeloid stroma [12]. We already showed that N6L decreases tumour hypoxia of mPDAC and normalizes mPDAC vessels [24]. In this study, we demonstrated that the inhibition of nucleolin had an impact on both tumour vessels and CAFs, with a final result of reducing tumour hypoxia. These functional effects in the stroma have been further corroborated by RNA-Seq analysis, which identified a specific gene signature of activated and inflammatory stroma [8,9] induced by N6L, explaining at least in part the impairment of CAF proliferation and activation. Consistently, N6L inhibited IL-6 expression level in vitro and in vivo, and the blockade of IL-6 using an anti-IL6 mAb decreased the frequency of Tregs and Gr-MDSCs, similar to N6L. IL-6 plays a central role in boosting immunosuppressive MDSC cells in the tumours, and MDSCs in turn are able to regulate Treg differentiation [5]. Our results suggest that the modulation of IL-6 secreted by CAFs could be part of the mechanism by which N6L inhibits the number of immunosuppressive cells in PDAC. Here again, and as previously mentioned for anti-LY6G treatment [43], anti-IL6 treatment recapitulates inhibition of Tregs and MDSCs as well as increased CD8^+^ T cell infiltration but not tumour growth inhibition. It has been shown that the lack of IL-6 in the tumour microenvironment of IL-6 deficient mice inhibited tumour growth of colon cancer cells [59]; however, blocking IL-6 by anti-IL-6 antibody alone did not block tumour growth of PDAC [60]. We propose that IL-6 inhibition could be part of the modalities of action of N6L. For instance, we could hypothesize that the tumour vessel normalization effects of N6L could be in part responsible for increasing the amount of CD8^+^ T cell infiltration, such as anti-VEGF treatment or dual inhibition of VEGF and Ang-2 [36,61]. However, while low-dose anti-VEGFR2 therapy in orthotopic mPDAC impacts the structure of tumour vessels by enhancing the pericyte coverage, as previously described in other tumour types [40,62], differently from N6L, low-dose anti-VEGFR2 was not able to impact the proportion of Tregs or MDSCs. All these data suggest that N6L can exert a pleiotropic effect on the tumour microenvironment and tumour cells in PDAC, compared to anti-IL-6 and other anti-angiogenic drugs. We could therefore envisage that the observed re-programming of immune cells induced by N6L could mainly be due to a dual effect on CAFs and tumour vessels.

In conclusion, this work highlights a new therapeutic strategy that restrains immunosuppressive cells of the PDAC microenvironment and promotes T cell infiltration and activation. We elucidated a new mechanism of action of the anti-nucleolin therapy that targets CAFs activation and expansion. This additional and novel effect is complementary to the direct impact of N6L on tumour cell growth and on tumour vessels, making N6L a tumour microenvironment–normalizing drug.

## Figures and Tables

**Figure 1 cancers-14-04265-f001:**
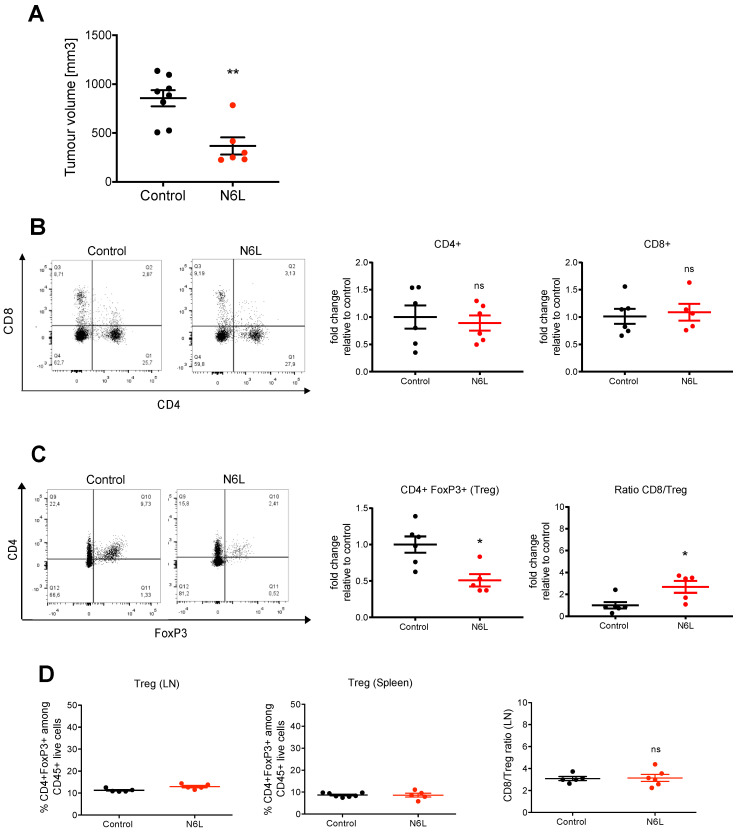
N6L reduces the amount of regulatory T lymphocytes in mPDAC tumours. Immunocompetent syngenic FVB/n mice were injected with mPDAC cells into the pancreas. Mice were treated one week after cell inoculation with N6L (7 mg/kg) or saline solution by i.p. injections three times a week for three weeks. (**A**) Mice were sacrificed, and tumour volumes were measured. Graph shows a representative experiment (from three independent experiments). (**B**,**C**) CD4^+^, CD8^+^ T lymphocytes and regulatory T lymphocytes (Tregs) from tumour tissues were analysed by flow cytometry among the CD45^+^ cells, and the fold change of the % of CD45^+^CD4^+^, CD45^+^CD8^+^, or CD45^+^CD4^+^FoxP3^+^ cells of N6L-treated tumours relative to the control are plotted in the graphs. Graphs show the fold changes of three experiments. (**D**) The graphs show the % of CD45^+^CD4^+^FoxP3^+^ of control and N6L-treated mice in lymph nodes and in spleen of mice of the representative experiment in (**A**), and the CD8/Treg ratio in lymph nodes. In A (Control *n* = 8 mice, N6L *n* = 6 mice) and in B, C and D the sample number is indicated on the graphs and depends on the efficiency of cell isolation and flow cytometer analysis. *p*-values were calculated between indicated conditions by the two-tailed Mann-Whitney U-test (**, *p* < 0.01; *, *p* < 0.05; ns = not significant).

**Figure 2 cancers-14-04265-f002:**
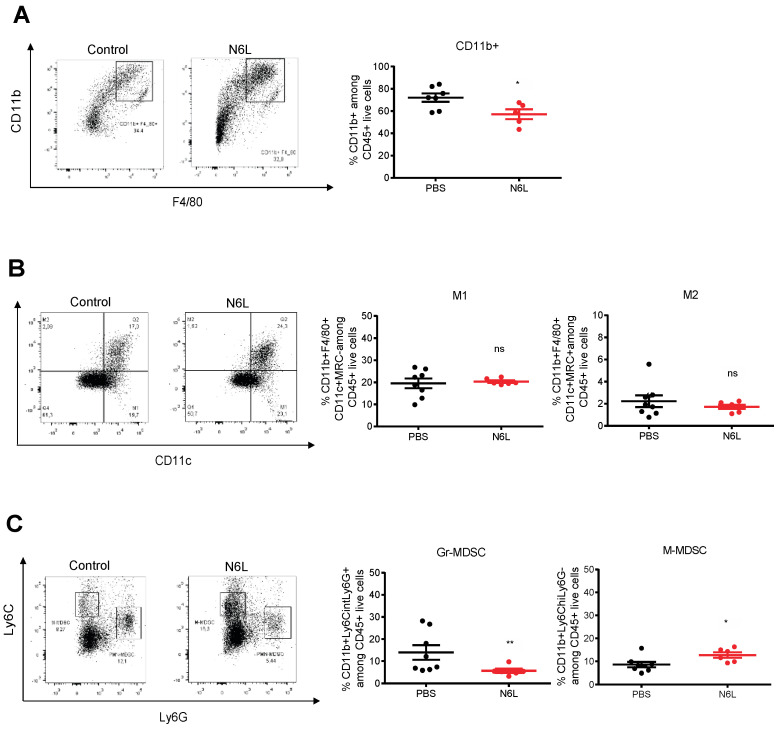
N6L impacts myeloid cell infiltration in mPDAC tumours. Myeloid cells infiltrated in control or N6L-treated tumours were analysed by flow cytometry. (**A**) Frequency of CD11b^+^ myeloid cells of control and N6L-treated tumours. (**B**) Tumour-associated macrophages were analysed by flow cytometry and the % of M1 (CD45^+^CD11b^+^F4/80^+^CD11c^+^MRC^-^) and of M2 (CD45^+^CD11b^+^F4/80^+^CD11c^−^MRC^+^) macrophages of control and N6L-treated tumours were plotted as indicated. (**C**) Myeloid suppressor cells (MDSCs) were analysed by flow cytometry, and the % of Gr-MDSCs (CD11b^+^Ly6G^+^Ly6C^low^) and of M-MDSCs (CD11b^+^Ly6G^−^Ly6C^hi^) were plotted as indicated. *p*-values were calculated between indicated conditions by two-tailed Mann-Whitney U-test (**, *p* < 0.01; *, *p* < 0.05; ns = not significant; n are indicated on graphs).

**Figure 3 cancers-14-04265-f003:**
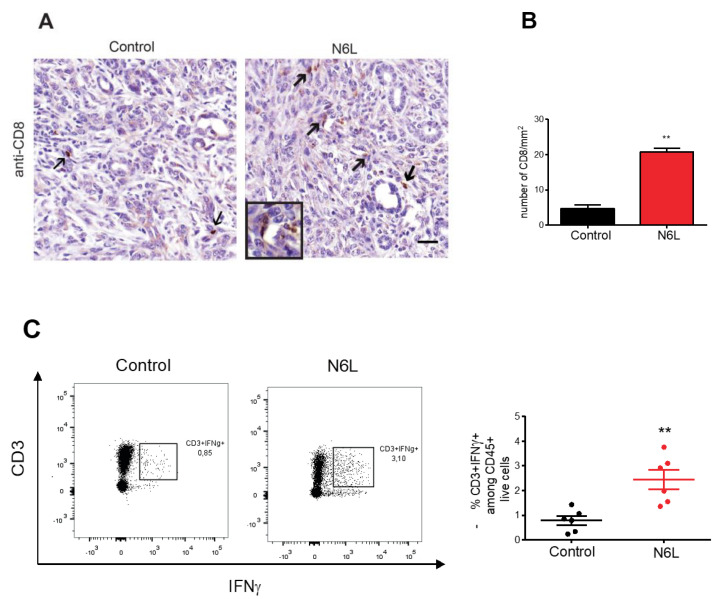
N6L increases lymphocyte infiltration and activation in mPDAC tumours. (**A**) Tumour sections of control and N6L-treated mice were immunostained by an anti-CD8 antibody to detect CD8^+^ cells (arrows); scale bar: 50 μm. (**B**) CD8^+^ cells in tumoral regions (at least 4) of tumour slices were counted by using QuPath software, and the results were plotted as a mean for each tumour (*n* = 6) as the number of cells/mm^2^. Two-tailed Mann-Whitney U-test (**, *p* < 0.01; *n* = 5 tumours) was applied. (**C**) Effector T cell activation was analysed by flow cytometry, and the % of CD3^+^IFNγ^+^ cells in control and N6L-treated mice is plotted (two-tailed Mann-Whitney U-test **, *p* < 0.01; *n* = 6 mice).

**Figure 4 cancers-14-04265-f004:**
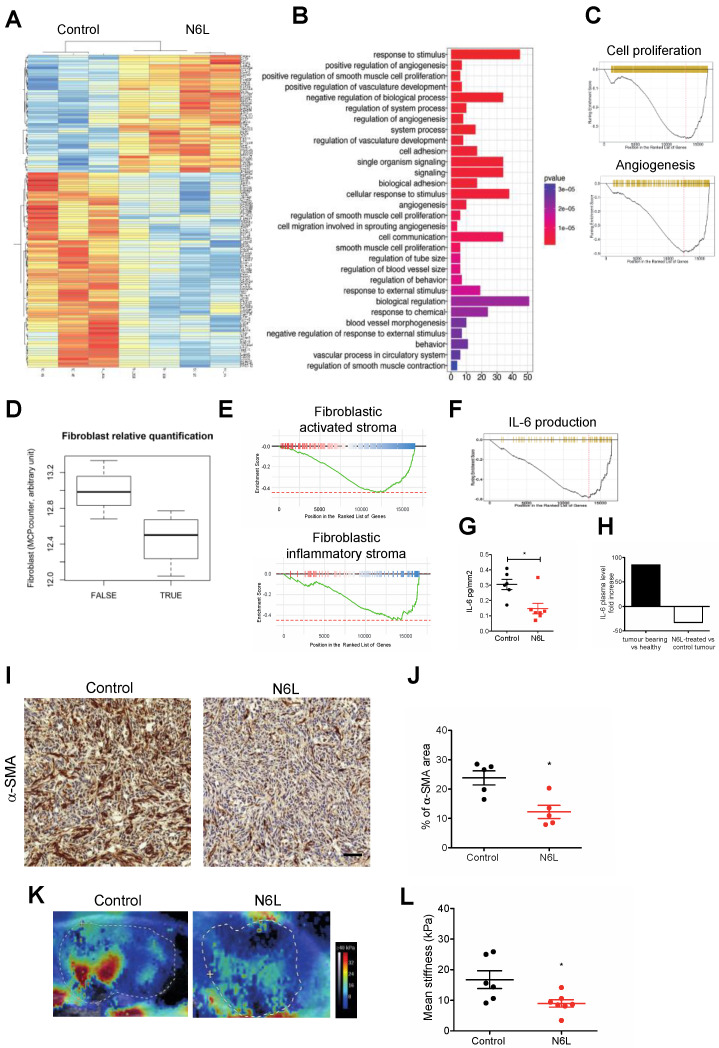
N6L regulates tumour stroma compartment. Tumours from mPDAC mice were harvested, and mRNAs were analysed by RNAseq. Data were normalized using the DESeq method. The heatmap of the differentially expressed genes is shown in (**A**); the Gene Ontology enrichment analysis is represented in (**B**), where the bars represent the number of genes and the colour, the *p*-value of the Fisher exact test; selected results of the GSEA analysis of regulated genes are shown in (**C**,**E**,**F**). (**D**) Boxplot of the RNA-based quantification of fibroblasts using MCP counter comparing N6L treatment with control. (**G**) IL-6 protein level in the tumours of control and N6L-treated mice quantified by ELISA are shown as the pg/mm^2^ of tumour (two-tailed Mann-Whitney U-test, *, *p* < 0.05, control *n* = 6, and N6L *n* = 7 tumours). (**H**) IL-6 semi-quantitative analysis of the plasma protein level from a pool of distinct cohorts of mice (healthy mice, tumour-bearing control mice, tumour-bearing mice treated by N6L), performed by protein array. Fold increase of healthy mice vs. tumour-bearing control mice and tumour-bearing N6L-treated mice vs. tumour-bearing control mice are represented. (**I**) Tumour sections of control and N6L-treated mice were immunostained by an anti-α-SMA antibody; the area covered by CAFs expressing anti-α-SMA was quantified and results are plotted in (**J**) (two-tailed Mann-Whitney U-test, *, *p* < 0.05; *n* = 5 tumours). Scale bar: 50 μm. (**K**,**L**) Tumour stiffness was measured using shear wave elastography (SWE), as described in Materials and Methods, in control and N6L-treated tumours at the endpoint of the experiment (two-tailed Mann-Whitney U-test, *, *p* < 0.05, *n* = 6 tumours).

**Figure 5 cancers-14-04265-f005:**
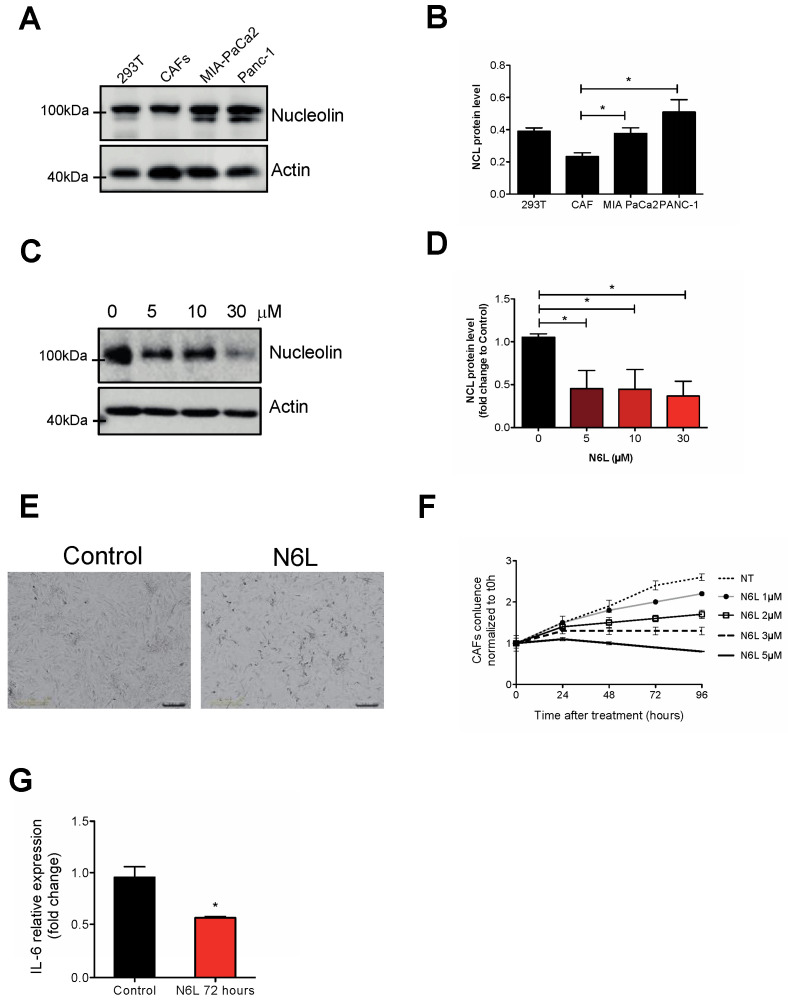
N6L regulates CAF viability and IL-6 expression. (**A**) Immunoblot analysis of nucleolin in cell lysates from HEK293T (293T), human immortalized CAFs, MIA PaCa-2, and PANC-1 cells or (**C**) of nucleolin in CAFs treated or not with N6L (5, 10, or 30 µM) for 24 h. β-Actin was used as loading control. Nucleolin protein level of Western blots was analysed, and the ratio of nucleolin/β-actin intensity in different cell lines is shown in (**B**) or after N6L treatment as fold change to untreated cells 0 (**D**). (**C**,**D**) CAFs were treated by increasing concentrations of N6L (1, 2, 3, and 5 µM), and cell growth was assessed by Incucyte^®^ live-cell imaging over 96 h. Representative pictures of 5 µM N6L and control are shown in **C**. Scale bar: 100 μm. (**D**) CAF confluence (%) was calculated with Incucyte^®^ software and normalized to non-treated cells. (**E**,**F**) IL-6 relative expression analysis in CAFs non-treated or treated with N6L 5 µM for 72 h. mRNA levels were normalized to GAPDH. For (**B**,**D**,**G**) (two-tailed Mann-Whitney t-test, *, *p* < 0.05, *n* = 3 experiments).Uncropped WB images were shown in Appendix A.

**Figure 6 cancers-14-04265-f006:**
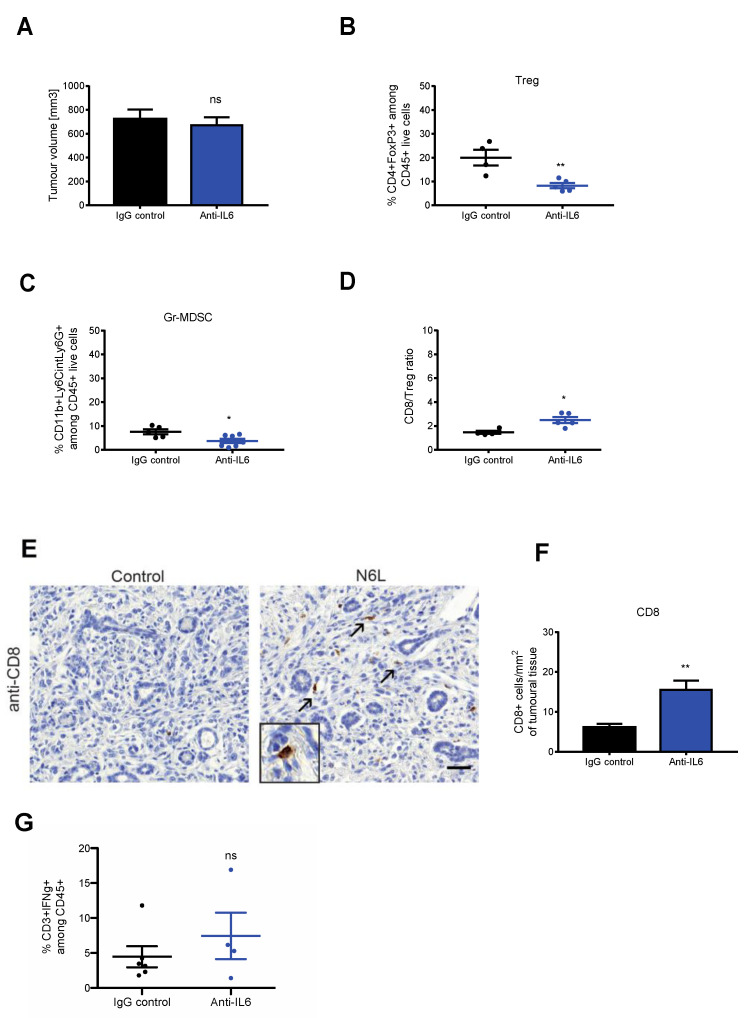
Anti-IL-6 antibody mimics N6L effects on PDAC immune microenvironment. mPDAC tumours were generated as in Figure 2 and treated with anti-IL-6 blocking antibody or IgG control antibody for three weeks, three times per week, by i.p. injection. Mice were sacrificed 21 days after the treatments. (**A**) Tumour volumes were measured as in Figure 2, and graphs show a representative experiment (from two independent experiments). (**B**–**D**) Immune cell populations in tumour tissues were analysed by flow cytometry as in Figure 2. Tregs and PMN-MDSC were analysed, and the results are plotted as the fold change of the % of (**B**) CD45^+^CD4^+^FoxP3^+^ cells and (**C**) CD11b^+^Ly6G^+^Ly6C^low^ cells in N6L-treated tumours relative to control tumours. (**D**) Fold change of the CD8/Treg ratio between control and N6L-treated tumours. (**E**) Tumour sections of control and anti-IL-6-treated mice were immunostained by an anti-CD8 antibody, CD8^+^ cells were counted (arrows), and results were plotted as (**F**) the number of cells/mm^2^. Scale bar: 50 μm. *p*-values were calculated between indicated conditions by two-tailed Mann-Whitney *t*-test (**, *p* < 0.01; *, *p* < 0.05; *n* = 5 mice). (**G**) Effector T cell activation was analysed by flow cytometry as in Figure 3C, and the % of CD3^+^IFNγ^+^ cells in control and N6L-treated mice is plotted (two-tailed Mann-Whitney U-test **, *p* < 0.01; control, *n* = 5 mice; anti-IL-6, *n* = 4 mice).

## Data Availability

The data presented in this study are available in this article and Appendix A.

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
