# Peer review of "Nucleolin Therapeutic Targeting Decreases Pancreatic Cancer Immunosuppression"

_cancers, 2022, doi:10.3390/cancers14174265_

Round 1

Reviewer 1 Report

This article is well documented with significant set of experiments. I have little concern before accepting this for publishing in Cancers:

1.     Line no 88: hav:e to be replaced by have

2.     Line no 116 to 120: As only the 1000 or 10,000 cells were injected per mice, how authors had determined the tumor burden or confirmed the growth of tumor cells one week after cells injection into mice before the N6L or other antibody treatment? 

3.     Line no 177-178: Please explain why there is a huge time gap (1 hours to 15 hours) for fixing cells with the FoxP3 transcription kit?

4.     Appropriate labeling of figures at various places are missing, for example supplementary figure 1A, this section is from the tumor section of PDAC etc, Figure Supplementary 3A which one is treated or control group?   

5.     The Catalogue number of each chemical/protein array etc used in this manuscript should be mentioned.

6.     What was the source of CAF in figure 5? Were these isolated from PDAC tumors? 

7.     Line 423 to 425: This represents that mice were treated with anti-IL6 antibody and have impact on CD8 T cells infiltration whereas in Figure 6E and figure legend 6E, it is mentioned about treatment with N6L. Why is there a discrepancy? 

8.     Discuss why there is no change in tumor volume after the anti-IL 6 treatment in mice? What was the status of the activation marker of T cells after the anti-IL-6 treatment? 

9.     Combining anti-IL6 and N6L or N6L along with current standard treatment plan for PDAC could have a better synergistic impact on reducing the tumor volume and more significant changes in suppressive tumor immune microenvironment. Were the authors thought about this treatment plan in their PDAC model? 

Author Response

Reviewer 1

This article is well documented with significant set of experiments. I have little concern before accepting this for publishing in Cancers:

  1. Line no 88: hav:e to be replaced by have

We corrected, thank you

  1. Line no 116 to 120: As only the 1000 or 10,000 cells were injected per mice, how authors had determined the tumor burden or confirmed the growth of tumor cells one week after cells injection into mice before the N6L or other antibody treatment? 

By ad hoc studies we have previously defined the starting point in order to perform a regression trial one week after cancer cells inoculation (as previously described, Gilles, CR 2016). In this time-period PDAC tumors reached a volume of approximately 80mm3. This information has been added in Material and Methods

  1. Line no 177-178: Please explain why there is a huge time gap (1 hours to 15 hours) for fixing cells with the FoxP3 transcription kit?

We corrected in in Material and Methods: 1 hour or over night (O.N.)

  1. Appropriate labeling of figures at various places are missing, for example supplementary figure 1A, this section is from the tumor section of PDAC etc, Figure Supplementary 3A which one is treated or control group?   

We have changed and modified the labeling of the figures as shown on the new version of the manuscript

  1. The Catalogue number of each chemical/protein array etc used in this manuscript should be mentioned.

We added the catalogue number of each chemical/protein array used

  1. What was the source of CAF in figure 5? Were these isolated from PDAC tumors? 

We have better described In Material and Method as following: Immortalized human CAFs were cultured in in Advanced DMEM/F12 (Thermo 12634) 10% FBS, at 37°C 5% CO2, as previously described (39).        And in

results:

  1. Line 423 to 425: This represents that mice were treated with anti-IL6 antibody and have impact on CD8 T cells infiltration whereas in Figure 6E and figure legend 6E, it is mentioned about treatment with N6L. Why is there a discrepancy? 

         Indeed, is not correct and there was a typing mistake. We modified and changed accordingly.

  1. Discuss why there is no change in tumor volume after the anti-IL-6 treatment in mice? What was the status of the activation marker of T cells after the anti-IL-6 treatment? 

         It has been shown that IL-6 lack in the tumour microenvironment of IL-6          deficient mice          inhibited tumour growth of colon cancer cells (Ohno et al.   2017)), however blocking            IL-6 by anti-IL-6 antibody alone did not block           tumour growth of PDAC (Mace et al.   2018). We therefore reproduce the        data of Mace et al. We have added this sentence     to the discussion section.

            Moreover, we could add in Figure 6 the results of the CD3+ expressing IFN      gamma after treatment of anti-IL-6. In Figure 3 we showed that N6L     increased the frequencey            of CD3+IFNgamma+ cells while in the new Figure 6            we show that the anti-IL-6 was not sufficient to increase this frequency.   Together with the published targeting of tumour        cells of N6L, this result      may explain why N6L is able to delay tumour growth          compared to anti-IL-6.

            We have added to the discussion the following sentence « All these data          suggest that             N6L can exert a pleiotropic effect on tumour microenvironment     and tumour cells in            PDAC, compared to anti-IL-6 and other anti-angiogenic     drugs ».

  1. 9. Combining anti-IL6 and N6L or N6L along with current standard treatment plan for PDAC could have a better synergistic impact on   reducing the tumor volume and more significant changes in suppressive tumor immune microenvironment. Were the authors thought about this      treatment plan in their PDAC model?

         We thank you the reviewer for this question. Concerning the          combination of N6L with standard of care treatment in the previous         paper Maud et al. Cancer Res 2016, we could show a synergistic effect   of N6L+gemcitabine treatments.

         Here, in this manuscript, we did not plan a combination of N6L+anti-IL6 since both inhibit immunosuppressive cells and we would possibly         expect a not significant synergistic effect in reducing tumor growth.         However, we planned and tried a combination of N6L with an     immunocheckpoint inhibitor that we thought could target both        immunosuppressive cells (Treg and MDSC) and immunosuppressive         signals (PD1/DL1 axis). In a pilot trial, the N6L inhibited tumour growth,   the anti-PD1 did not have any effect and the combination of N6L and        anti-PD1 inhibited tumour growth at the same extent of N6L suggesting that the anti-tumoral effect of the combo was due to the N6L and that the        combination of the 2 drugs did not have synergistic effects.

         We are currently investigating different combinatorial strategies to          improve the efficacy of immunotherapeutic approaches in our PDAC        models. For instance, we are planning to combine N6L with other   immunocheckpoint inhibitors such as anti PD-L1 or other modulators of          immune-suppressive myeloid cells (such as anti- CD40). This PDAC       model has a very aggressive phenotype and we believe, in line with          recent findings describing the immunotherapy in PDAC, that the    combination of more than one immuno-check point drug (that targets      diverse immune cell populations) with N6L would have a greater effect in reducing tumor progression.

         At the moment, due to the complexity and different arms of treatments, we do not have data available on these different combinatorial       strategies.

Reviewer 2 Report

The article adressess an “hot” topic regarding PDAC microenvironment and its relantionship with immune system. PDAC is considered resistant to many chemoterapeutic drugs and immune checkpoint inhibitors, and this study investigated and explained some of the mechanisms involved in this resistance using a mouse model of PDAC. The study provides evidence that immune environment of PDAC can be modulated and made less immunosuppressive in part by reduction of CAFs and IL6 production which are associated with reduced proportion of Tregs and myeloid-derived immunosuppressive cells and increased number of IFN-gamma producing effector T cells.

Although the immunomodulatory effect does not seem to be directly responsibile for the antiproliferative activity observed upon treatment with N6L in this mouse model, the evidence that PDAC immune environment may be modulated and even straightened as the Authors wrote, represents an important achievement in PDAC scientific research. Of course it is crucial to verify if this effect of N6L can be reproduced also in PDAC patients.

The only correction I suggest is the following: in the first paragraph (3.1) of Results at line 261, if the proportion of Treg cells increased in tumor bearing mice compared to healthy mice then there would be a DECREASE of the CD8+/Treg ratio and not a 2.3 fold INCREASE of the ratio as written.

Author Response

Reviewer 2

The article adressess an “hot” topic regarding PDAC microenvironment and its relantionship with immune system. PDAC is considered resistant to many chemoterapeutic drugs and immune checkpoint inhibitors, and this study investigated and explained some of the mechanisms involved in this resistance using a mouse model of PDAC. The study provides evidence that immune environment of PDAC can be modulated and made less immunosuppressive in part by reduction of CAFs and IL6 production which are associated with reduced proportion of Tregs and myeloid-derived immunosuppressive cells and increased number of IFN-gamma producing effector T cells.

Although the immunomodulatory effect does not seem to be directly responsibile for the antiproliferative activity observed upon treatment with N6L in this mouse model, the evidence that PDAC immune environment may be modulated and even straightened as the Authors wrote, represents an important achievement in PDAC scientific research. Of course it is crucial to verify if this effect of N6L can be reproduced also in PDAC patients.

The only correction I suggest is the following: in the first paragraph (3.1) of Results at line 261, if the proportion of Treg cells increased in tumor bearing mice compared to healthy mice then there would be a DECREASE of the CD8+/Treg ratio and not a 2.3 fold INCREASE of the ratio as written.

We thank you a lot the reviewer. The specific aspect of the results at line 261 was effectively an error. We have now corrected it in the revised version of the manuscript.

Reviewer 3 Report

The manuscript titled “Nucleolin therapeutic targeting decreases pancreatic cancer immunosuppression” describes pseudopeptide N6L restrains immunosuppressive cells of PDAC microenvironment and promotes T cells activation impact on tumor growth and angiogenesis, which suggested N6L might be a potential drug of cancer therapy. The followings are some concerns and comments have been pointed out that the authors may want to consider.

1) Line 97 Materials and Methods: a) Please provide more details for the reagents to make your work relatively easier reproducible. b) I’d suggest the authors include some more N6L information in the manuscript, for example, sequence, if possible.

2) Line 107: Cell number should include.

3) Line 116: Please correct the unit. And check throughout the manuscript.

4) Line 199: Please be consistent with “+” as superscript.

5) Line 244: Figure 1: a) Had all the mice samples been tested? Please explain why the sample size does not match? b) Please include the  final day for the tumor volume.

6) Line 255: I’d suggest the authors use italic p as it refers to a p-value. Check throughout the manuscript.

7) Line 261: It should be “2.3 folds”.

8) Line 290 Figure 2: The sample size in the figure legend does not match with figure images. Please explain.

9) Line 298 “n=6”, line 327 “n = 6”, please consistent with or without a space before and after the signs; check throughout the manuscript;

10) Line 321 Figure 3: Figure 3C X-axis should be “IFN” instead of “INF”.

11) Line 323: Please correct the scale bar unit.

12) Line 341: Figure 4B, please provide detailed information on “gene onthology” and “p-value” analysis in the methods section.

13) Line 356: The authors stated n=6; Please explain why Figure 4G, N6L group n=7, Figure 4J, control group N6L group n=5, and Figure 4L, N6L group n=7?

14) Figure 3A, Figure 6E: Please remove “white background” from the “arrows”.

15) Except in Figure 4B, are there any solid data to support N6L effects on angiogenesis? Please provide. For example, staining or whatever else.

16) Please delete extra spaces throughout the manuscript.

17) The reviewer would highly suggest the authors remove unnecessary self-citations.

Author Response

Reviewer 3

The manuscript titled “Nucleolin therapeutic targeting decreases pancreatic cancer immunosuppression” describes pseudopeptide N6L restrains immunosuppressive cells of PDAC microenvironment and promotes T cells activation impact on tumor growth and angiogenesis, which suggested N6L might be a potential drug of cancer therapy. The followings are some concerns and comments have been pointed out that the authors may want to consider.

  1. Line 97 Materials and Methods: a) Please provide more details for the reagents to make your work relatively easier reproducible. b) I’d suggest the authors include some more N6L information in the manuscript, for     example, sequence, if possible.

         We have added the information requested in Materials and Methods

  1. Line 107: Cell number should include.

         We have added the cell number

  1. Line 116: Please correct the unit. And check throughout the manuscript.

         We corrected and check throughout the manuscript

  1. Line 199: Please be consistent with “+” as superscript.

         We modified accordingly

  1. Line 244: Figure 1: a) Had all the mice samples been tested? Please explain why the sample size does not match? b) Please include the  final day for the tumor volume.

            Yes, all tumours were tested in flow cytometer analysis. However, in A (Control n=8 mice, N6L n=6 mice) in B, C and D the isolation of some sample             did give enough cells for flow cytometer analysis and n is indicated in each     graph. I have added this sentence in the Figure legend

  1. Line 255: I’d suggest the authors use italic p as it refers to a p-value. Check throughout the manuscript.

         We corrected accordingly

  1. Line 261: It should be “2.3 folds”.

         Here there was an error that reviewer 2 noticed. Now the sentence is:

            However, compared to healthy mice, the proportion of Treg cells significantly increased in tumour bearing mice leading to a 2.3 fold decrease of the           CD8+/Treg ratio.

  1. Line 290 Figure 2: The sample size in the figure legend does not match with figure images. Please explain.

         Indeed, there was a mistake, the Control n=8 mice, N6L n=6 mice as in       Figure 1 but the number of sample in graphs is dependent on the isolation and analysis of the flow cytometry. We added and modified in Figure 2 legend that            n are indicated on graphs.

  1. Line 298 “n=6”, line 327 “n = 6”, please consistent with or without a space before and after the signs; check throughout the manuscript;

         We corrected throughout the manuscript

  1. Line 321 Figure 3: Figure 3C X-axis should be “IFN” instead of “INF”.

         We corrected

  1. Line 323: Please correct the scale bar unit.

         We corrected

  1. Line 341: Figure 4B, please provide detailed information on “gene onthology” and “p-value” analysis in the methods section.

We have added the detailed information in methods and the Figure legend.

  1. Line 356: The authors stated n=6; Please explain why Figure 4G, N6L group n=7, Figure 4J, control group N6L group n=5, and Figure 4L, N6L group n=7?

We thank you the reviewer for the correction. In Figure 4 the trial of G, H and J panels are the same but for immunohistochemistry analysis we analysed only n=5 tumours. Figure 4L was an independent trial with n=6 tumours. We have now better explained the groups in the Figure legend of the revised manuscript.

  1. Figure 3A, Figure 6E: Please remove “white background” from the “arrows”.

I am sorry, I do not understand this point.Could you please explain more in detail which changes we have to make?

  1. Except in Figure 4B, are there any solid data to support N6L effects on angiogenesis? Please provide. For example, staining or whatever else.

We thank the reviewer for this question. In Gilles et al. Cancer Res 2016 we showed by immunofluorescence of frozen tumours that vessel density and vessel branching were significantly decreased in N6L-treated mPDAC and that remaining tumor blood vessels showed an increase of pericyte coverage. These effects were accompanied by an increase of vessel perfusion and drug delivery and a decrease of tumour hypoxia. All these results allowed us to conclude that N6L normalizes tumour vessels in PDAC. In the paper submitted to Cancers the aim was to study the effects of N6L on immune cells infiltrated in tumours. All tumours were used for flow cytometer analysis or immunohistochemistry of paraffine-embedded sections of tumours. The effect of N6L on tumour angiogenesis treated for immune cell analysis was further confirmed and corroborated by RNAseq showed in Fig4B.

  1. Please delete extra spaces throughout the manuscript.

 We corrected accordingly

  1. The reviewer would highly suggest the authors remove unnecessary self-citations.

All citations were justified by the fact that our laboratory patented the pseudopeptide N6L and therefore all papers published with this drug have researcher of the lab as authors or co-authors. However, we decided to remove:

         - reference number 26 because it concerns the anti-angiogenic activity   of N6L in retinopathy and that we already cited the anti-angiogenic     activity in cancer;

         - reference number 33 because it concern the activity of N6L in synergy          with a toxin.

Reviewer 4 Report

Authors need to check grammar errors with the special characters. Also correct the size font of the figure legends, as many of them have a different size. 

I will recommend accepting this article in its present form after a grammar check. 

Author Response

Reviewer 4

Authors need to check grammar errors with the special characters. Also correct the size font of the figure legends, as many of them have a different size. 

I will recommend accepting this article in its present form after a grammar check. 

Thank you to the reviewer for his/her comments. We checked grammar errors and modified the size font of the figure as requested. The changes are now in the new version of the revised manuscript

Round 2

Reviewer 3 Report

Thank you for the revised manuscript. Second-round comments as below.

1) First-round comments 6: I’d suggest the authors please be consistent with italic p as it refers to a p-value.

2) First-round comments 14: Figure 3A, the authors used “arrows” to point to “anti-CD8” positive cells. The “arrows” in the insert squares were filled with a “white background”.  I’d suggest the authors use “no fill” to make it clearer.

3) Please modify the citation format according to the /cancers/ guideline to the authors, [] for citation.

Author Response

We thank the reviewer for his suggestions.

We have changed the p-value in p-value.

We have changed the arrows in Figure 3A

We have corrected the reference style
